# Resource Scheduling and Energy Consumption Optimization Based on Lyapunov Optimization in Fog Computing

**DOI:** 10.3390/s22093527

**Published:** 2022-05-06

**Authors:** Chenbin Huang, Hui Wang, Lingguo Zeng, Ting Li

**Affiliations:** School of Mathematics and Computer Science, Zhejiang Normal University, Jinhua 321000, China; ryenchen1@163.com (C.H.); jk63@zjnu.cn (L.Z.); lianz_lit@zjnu.edu.cn (T.L.)

**Keywords:** IoT (Internet of Things), edge computing, fog computing, Lyapunov optimization

## Abstract

Delay-sensitive tasks account for an increasing proportion of all tasks on the Internet of Things (IoT). How to solve such problems has become a hot research topic. Delay-sensitive tasks scenarios include intelligent vehicles, unmanned aerial vehicles, industrial IoT, intelligent transportation, etc. More and more scenarios have delay requirements for tasks and simply reducing the delay of tasks is not enough. However, speeding up the processing speed of a task means increasing energy consumption, so we try to find a way to complete tasks on time with the lowest energy consumption. Hence, we propose a heuristic particle swarm optimization (PSO) algorithm based on a Lyapunov framework (LPSO). Since task duration and queue stability are guaranteed, a balance is achieved between the computational energy consumption of the IoT nodes, the transmission energy consumption and the fog node computing energy consumption, so that tasks can be completed with minimum energy consumption. Compared with the original PSO algorithm and the greedy algorithm, the performance of our LPSO algorithm is significantly improved.

## 1. Introduction

With the evolution of technology, people’s lives are surrounded by a greater number of IoT devices. Data in IoT are also growing exponentially. If all this traffic is handed over to the cloud, the computing pressure on the cloud will be enormous [1]. Therefore, fog computing has arisen, first, to use the idle resources of IoT devices, second, to reduce the flow of cloud center network, and third, to reduce the delay of tasks. So far, fog network has been widely applied, including to the Internet of vehicles, smart buildings, smart cities, and industrial automation. A large number of application scenarios produce a large number of task requirements [2,3,4]. Among them, delay-sensitive tasks pose a great challenge to the computing capability of nodes. If the task cannot be completed within the specified time, it will be meaningless even if the task is completed. Therefore, the task needs to be offloaded into the fog network, and the nodes in the fog network can help the calculations [5].

The emergence of fog computing provides a solution to this kind of problem. Especially for delay-sensitive tasks, it can meet the needs of applications with high requirements on computing resources and delay, such as AR and online games. These are requirements that the cloud cannot solve, so we use fog networking to help cloud computing. A typical fog computing system consists of a set of fog nodes at different locations. These nodes are deployed on the periphery of the network and have flexible resource configurations such as storage, computing, and network bandwidth [6]. If the cloud is far from the edge node, the link connected to the cloud may not be reliable, which also causes a lot of delay. Therefore, in many cases, it does not make sense to put edge tasks in the cloud [7]. Hence, fog computing maximizes efficiency by offloading and distributing the workload of terminal equipment and fog nodes, while minimizing energy consumption and ensuring delay.

Generally speaking, the energy consumption required for processing a task is the sum of the transmission time, transmission power, and processing capability of computing devices. In addition, the transmitted power combines channel state, such as the available bandwidth and data transfer rate. Thus, higher transmission rates can be achieved by increasing transmission power, which also reduces transmission time, but results in a greater energy consumption. Therefore, energy emissivity can be controlled by balancing transmission power and transmission time to achieve the lowest energy consumption while meeting the task delay requirement. The optimal task scheduling strategy should always control this balance and improve the transmission rate when the task delay requirement is high or reduce the transmission rate when the task delay requirement is low to achieve the energy-saving purpose [8].

In order to solve the above problems effectively, we use a Lyapunov optimization method and Lyapunov drift theory. The reason for using this method is that constraint optimization of the mean can be achieved in a stochastic system without the need to estimate random events that will occur in the future. The method first lists the constraints in the system and then builds a virtual queue. To optimize the objective function in each time slot, a drift-plus-penalty weight parameter is introduced. The reason for adding weights is to use this parameter to control the balance between drift and optimization.

Although Lyapunov optimization function can guarantee the stability of virtual queue, this method is not suitable for the bandwidth limitation and computing resource limitation of task offloading and processing in a fog network, because it has to be adapted to meet our needs. Therefore, we designed the PSO algorithm based on a Lyapunov optimization framework.

Our contributions to this paper are as follows:The concept of pro-node is proposed and our proposed LPSO algorithm is used to optimize it.A virtual queue is designed, and the optimization objective function is converted into the utility function under the Lyapunov framework. In addition, the balance between transmitting power and computing resource allocation can be achieved while ensuring queue stability and delay.We evaluate our proposed strategy by comparing it with different algorithms. In the case of tasks with delay constraints, our proposed method performs better in various QoS parameters, including energy consumption, computing resource, and channel output coefficient.

The following parts of this paper are organized as follows: Section 2 introduces related work. Section 3 introduces our system model and the process of problem formulation. Section 4 mainly analyzes the problem, transforms it into a Lyapunov optimization problem and calculates its upper bound. Section 5 shows the algorithm we designed. Section 6 mainly compares the performance of LPSO with other algorithms. Finally, Section 7 is our conclusion and some expectations for future work.

## 2. Related Work

In recent years, many people have been interested in fog computing and edge computing and have also done a lot of related research. Ramadan et al. [9] proposed an energy-saving framework for fog networks based on multireceiver wireless sensor networks. They used two greedy algorithms to solve the multireceiver deployment problem of Harris hawks optimizer (Hho) and modified Harris hawks optimizer (Mhho), introducing the connection problem of wireless sensor networks (WSN) with fog network through a set of sink nodes (SNs). Arshed et al. proposed a resource-aware scheduler to assign incoming application modules to fog devices by placing application modules from the fog layer to the cloud instead of sending them directly to the cloud. Their placement strategy reduced the cost of using cloud resources with minimal execution time and minimal bandwidth usage [10]. Lin et al. [11] proposed a mathematical model for the multiterminal computing offloading problem, optimized the long-term average response delay objective under the constraints of long-term computing and energy consumption, and used the perturbation technique of the Lyapunov optimization method to convert the original problem into a deterministic cap problem. Moreover, a distributed optimization algorithm was used to solve the upper bound problem per time slot, where each IoT device exchanges a limited amount of information with its associated edge nodes and decides on its own for resource allocation and computing offloading.

Liao et al. [12] used Directed Acyclic Graph (DAG) technology in the edge computing scheme to model the scheduling problem of application-dependent mobile tasks in the application of the allocation and scheduling of complex application-dependent applications and proposed a priority quantification method for application-dependent applications, considering the data volume, effort, deadlines, and dependencies. The estimated priorities were used as heuristics for task assignment and scheduling, effectively solving the NP-hard joint optimization problem.

Wang et al. [13] proposed a computing offloading architecture consisting of multiple fog nodes with heterogeneous communication and computational capabilities and a cloud center. They model dynamic communication and computing resources as two dynamic queues for recording resource availability and improving resource utilization. The whole problem was decomposed into channel power control and task assignment problems. A modified branch-and-bound method was used to solve the task assignment problem without considering resource availability. Then, they checked the resource availability to refine it with the knapsack problem.

Ren et al. [14] optimized virtual placeholders for all queues in a distributed fashion to create sufficient queue variance and activated Lyapunov optimizations under an infinite buffer setting. They proved that the optimal virtual placeholder was a new three-layer shortest-path problem and solved it by the Bellman–Ford algorithm.

Chang et al. [15] proposed a dynamic optimization scheme for an IoT fog computing system with multiple Mobile Devices (MDs) and utilized Lyapunov optimization to design an online dynamic algorithm to solve the joint optimization problem between a mobile device with harvesting energy and a fog node. Specifically, with the objective to minimize the system overhead related to delay, energy consumption, they proposed a joint computation offloading and channel resource allocation algorithm based on Lyapunov optimization, through minimizing the derived upper bound of the Lyapunov drift-plus-penalty function and dividing the main problem into several subproblems at each time slot and addressing them accordingly.

Mukherjee et al. [16] studied the optimum scheduling policy of the two-queue system in a fog node that allows a higher number of deadline-aware offloaded tasks to be processed while making both queues stable. The proposed approach was decoupled into two strategies: the priority-aware scheduling policy applying the Lyapunov drift-plus-penalty function and the fog nodes collaboration based on each fog node’s queueing status under the scheduling policy. The authors could effectively solve the problem of task completion rate.

Zhang et al. [17] studied resource scheduling for delay minimization in multiserver cellular edge computing systems. The traditional method defines queue-length-based Lyapunov functions and designs scheduling algorithms which solve the corresponding queue length control problem.

The experiments of Ren et al. were more ideal, as the buffer was not infinite, and they also reduced the delay but did not take into account the optimization of delay-sensitive tasks and the grade of completion of delay-sensitive tasks. In the research of Chang et al., only the energy consumption was optimized without considering the limit of task delay. Mukherjee et al. did not consider the energy consumption. In this paper, we study the combination of channel power and computing resource issues in edge computing networks, where multiple end-users (IoT devices) are considered to generate variable traffic loads. The goal is to minimize long-term system execution overhead when channel resources, computing resources, and task execution time are all limited. By extending the Lyapunov optimization, we transform the original problem into an upper-bound problem and design a scalable distributed algorithm. The symbols used in the paper are defined in Table 1.

## 3. System Model

As shown in Figure 1, our system is modeled by a set *N* = {1, 2, …, *N_i_*} consisting of *N* IoT nodes, and a set *M* = {1, 2, …, *M_j_*} consisting of *M* fog nodes, and pro-nodes connected to the fog nodes. Tasks are randomly generated at each IoT node, and their arrivals are assumed to be independently and identically distributed. Then, in the long time case, the arrival of tasks can be considered as following the Poisson distribution. Therefore, we adopt M/M/1 as the upload queue model for the tasks. In this section, we propose a dynamic algorithm based on the Lyapunov optimization to solve the problem. The Lyapunov optimization is considered an efficient tool for designing online control algorithms without any prior knowledge [18,19,20]. However, since the power consumption is time-dependent, the decision sets in different time slots are not independently and identically distributed. Therefore, we advocate additional perturbation methods [19] to solve this problem.

### 3.1. Transmission Model

According to Shannon’s equation, the transmission rate of data transmitted from IoT nodes to fog nodes can be described as
(1)$$C_{i,j}(t)=W\log_{2}(1+\frac{P_{i}^{c}(t)S_{i,j}(t)}{WN_{0}})$$
where denote the transmission power,$S_{i,j}(t)$ is the channel gain, *W* denotes the bandwidth, *N*_0_ denotes the noise, and $C_{i,j}(t)$ denotes the transmission rate from *i* to *j* at time *t*. From the M/M/1 average delay formula [20], the uplink delay (including queuing and transmission time) can be given by the following equation
(2)$$\tau_{c}=\frac{1}{\frac{C_{i,j}(t)}{\alpha_{i}(t)D_{i}(t)}-\alpha_{i}(t)\lambda_{i}(t)}$$
where *τ_c_* denotes the delay of transmission, α is the ratio of offloading, and *D_i_* denotes the average data size of task *i*. Consider that if the total data size transmitted is constant and it is used to obtain the transmission rate, then it is not necessary to consider the length of an individual packet which does not affect the optimal decision and the optimal solution.

### 3.2. Energy Consumption Model

First is the energy consumption model for local processing, where the local task queue is in the M/M/1 model, and according to [20] the local computational delay $\tau_{i}$ can be expressed as the following equation:(3)$$\tau_{i}=\frac{\lambda_{i}(t)(\sigma^{2}F_{i}^{2}+{D_{i}}^{2})}{2(F_{i}^{2}-\lambda i(t)F_{i}D_{i})}+\frac{D_{i}}{F_{i}}$$

*λ_i_* denotes the arrival rate of task *i*, *σ* denotes the standard deviation of the task size, *F_i_* denotes the compute capability of *i*, and *D_i_* means the value of the task size for device *i*. Therefore, the total energy consumed, $P_{i}^{l}$, by performing its tasks is
(4)$$P_{i}^{l}=\sum_{i=1}^{N}P_{i}(t)$$
where $P_{i}(t)=\tau_{i}\varsigma {\left(F_{i}(t)\right)}^{3}$ means the energy consumed by IoT device *i*, *t* is the time slot, ς means a parameter that depends on the deployed hardware and is measurable in practice [18].

We divide the latency tasks into two categories. The first category of delay-sensitive tasks requires a latency less than *τ*_1_, and just a single node can satisfy its computation requirements, such as the local process in Figure 1.

The second category of delay-sensitive tasks requires a latency less than *τ*_2_, and the compute capability of one node alone cannot meet its latency requirements, so it needs to be offloaded to the pro-node and use the pro-node for joint computation, such as the offload in Figure 1.

Let the vector *τ* = [*τ*_1_, *τ*_2_].

For the first type of delay-sensitive tasks, its total energy consumption $P_{a1}(t)$ can be expressed as
(5)$$P_{a1}(t)=\sum_{j=1}^{M}P_{j}(t)+\sum_{i=1}^{N}P_{i,j}^{c}(t)+\sum_{i=1}^{N}\left(1-\alpha_{i}(t))P_{i}(t\right)$$
where $P_{j}(t)$ means the energy consumed by fog node *j* that can be calculated as $P_{j}(t)=\tau_{j}\varsigma {\left(F_{j}(t)\right)}^{3}$. $P_{i,j}^{c}(t)$ means the energy consumption of device *i* to transfer tasks to *j*. *M* denotes the set of IoT nodes, and *N* denotes the set of fog nodes.

For the second type of delay-sensitive tasks, its total energy consumption $P_{a2}(t)$ can be expressed as
(6)$$P_{a2}(t)=\sum_{s=1}^{S}P_{s}(t)+\sum_{i=1}^{N}P_{i,j}^{c}(t)+\sum_{i=1}^{N}\left(1-\alpha_{i}(t))P_{i}(t\right)$$
where $P_{s}(t)$ means the energy consumed by subnode *s* in the pro-node that can be expressed as $P_{s}(t)=\tau_{s}\varsigma {\left(F_{s}(t)\right)}^{3}$.

### 3.3. Problem Formulation

In this section, we optimize the total energy consumption $P_{a}(t)$ using the Lyapunov method. Here, we only prove that $P_{a1}(t)$ and $P_{a2}(t)$ can be obtained in the same way. Our objective is to minimize the energy consumption $P_{a1}(t)$. Here, we use $P_{a}(t)$ to express $P_{a1}(t)$. Therefore, the whole optimization problem can be transformed into a Lyapunov optimization problem, i.e., to ensure that the energy consumption of the average time slot is minimum:(7)$$P1:\;\min\underset{T\to \infty }{\lim}\frac{1}{T}\sum_{t=0}^{T-1}P_{a}(t)$$
(8)$$s.t.\underset{T\to \infty }{\lim}\frac{1}{T}\sum_{t=0}^{T-1}\sum_{i=1}^{N}F_{i,j}(t)<F_{j}$$
(9)$$\underset{T\to \infty }{\lim}\frac{1}{T}\sum_{t=0}^{T-1}\sum_{i=1}^{N}C_{i,j}(t)<C_{j}$$
(10)$$\;\underset{T\to \infty }{\lim}\frac{1}{T}\sum_{t=0}^{T-1}\tau (t)<\tau$$
(11)$$0\leq F_{ij}(t)\leq F_{j}$$
(12)$$0\leq C_{ij}(t)\leq C_{j}$$
(13)$$0\leq P_{i}(t)\leq P_{i}^{b}$$

The objective of P1 is to minimize the long-term average energy consumption of all IoT devices. The decision variables are the offload fraction $\alpha_{i}(t)$, the computing resource allocation $F_{i}$, the transmit power per time slot $P_{i}(t)$, and $F_{j}$ (or $F_{s}$), where *i* ∈ *N*. In P1, the decision variables *α* are influenced by the size of the tasks, the processing delay requirements, the channel bandwidth, the computational capability of $F_{i}$, $F_{j}$, and the transmit power of the IoT nodes. In this paper, we assume that the time slots are of the same length.

The constraint in P1 limits the computational capability allocated to the task from device *i* (8) and the transmission speed of device *i* (9) as well as the processing delay in the long term (10). In the long term, constrained by the computational capability of $F_{j}$, the computational capability allocated by $F_{j}$ to device *i* should be less than the computational capability of $F_{j}$ itself. The size of the channel bandwidth occupied by the task should be smaller than the total channel bandwidth in the long term, due to the limitation of the device *i*’s own transmission speed. If the task can be processed locally, then only the energy consumption of $F_{i}$ needs to be calculated.

## 4. Upper-Bound Analysis Based on Lyapunov Optimization

To solve P1, we convert it into a Lyapunov optimization problem and solve it using our proposed PSO algorithm based on the Lyapunov framework.

### 4.1. Setting up Virtual Queues

In Lyapunov optimization, the satisfaction of the long-term average constraint is equal to the rate stability of the virtual queue. Specifically, a virtual queue is provided to replace the computing resource constraint at the edge nodes (8), and *A*(*t*) denotes the random process of the length of the virtual queue *A* at time slot *t* [14]. *B*(*t*) and *C*(*t*), represent the channel constraint (9) and the computational delay constraint (10), respectively. Specific forms of *A*(*t*), *B*(*t*) and *C*(*t*) can be expressed as the following equations
(14a)$$A\left(t+1\right)=max\left(A\left(t\right)+\sum_{i=1}^{N}F_{i,j}(t)-F_{j},0\right)$$
(14b)$$B\left(t+1\right)=max\left(B\left(t\right)+\sum_{i=1}^{N}C_{i,j}(t)-C_{j},0\right)$$
(14c)$$C\left(t+1\right)=max\left(C\left(t\right)+\tau_{i}(t)+\tau_{c}(t)+\tau_{j}(t)-\tau ,0\right)$$

It is not difficult to find that in Equation (14a), if the request from *i* is larger than the computational capability of $F_{j}$, then queue *A* will get longer and longer and finally cause blocking, and vice versa, its queue will not increase. Queue *B*(*t* + 1) (14b) represents the next time slot of the *B*(*t*) state, i.e., the existing channel bandwidth occupation queue plus the channel bandwidth occupation of each node, then subtracting the total bandwidth yields the bandwidth occupation of the next time slot. Similarly to *B*(*t* + 1), *C*(*t* + 1) (14c) also represents the next time slot of the *C*(*t*) state, i.e., adding the upcoming delay to the existing delay queue, and then subtracting the delay requirements yields the delay queue of the next time slot. The initialization of virtual queues *A*, *B* and *C* consists in setting them to 0.

If virtual queue *A* is rate-stable, then by the definition of [19], only $\underset{T\to \infty }{\lim}A(t)/T=0$ can satisfy the restriction (8). Virtual queue *B* and virtual queue *C* can be obtained in the same way. The proof is as follows:

First, we prove that virtual queue *A* is stable:$$A\left(t+1\right)-A(t)=max\left(\sum_{i=1}^{N}F_{i,j}(t)-F_{j},-A\left(t\right)\right)$$

For t∈(0, 1, 2,…,T−1), there exists
$$\underset{T\to \infty }{\lim}\frac{A(T)-A(0)}{T}\geq \underset{T\to \infty }{\lim}\frac{1}{T}\sum_{t=0}^{T-1}\sum_{i=1}^{N}F_{i,j}(t)-F_{j}$$

If *A*(0) = 0, then we can find that
$$\underset{T\to \infty }{\lim}\frac{A(T)}{T}=0$$

Therefore, queue *A* is stable, and similarly, queue *B* and queue *C* are stable, so virtual queues *A*, *B*, and *C* can satisfy constraints (8), (9), and (10), respectively.

The proof is complete.

Using the above proof, we can convert P1 to P2:(15)$$P2:\;\min\underset{T\to \infty }{\lim}\frac{1}{T}\sum_{t=0}^{T-1}P_{a}(t)$$
(16)s.t. A(t) is rate stable
(17) B(t) is rate stable
(18) C(t) is rate stable

### 4.2. Setting the Drift-Plus-Penalty

We use the Lyapunov drift-plus-penalty [21] to solve for P2. According to [21], we first set up the virtual queueing vector.
(19)Θ(t)=[A(t),B(t),C(t)]
(20)$$\begin{array}{ll} L(\Theta (t)) & =\frac{1}{2}{\sum_{i=1}Q_{i}(t)}^{2}\\ & =\frac{1}{2}(A(t{)}^{2}+B(t{)}^{2}+C(t{)}^{2}) \end{array}$$
(21)ΔΘ(t)=E(L(Θ(t+1))−L(Θ(t))|Θ(t))

With stable rates across queues, P2 can then be converted to P3, because the original problem with a long-term average objective and constraints can be approximated to a problem with a drift-plus-penalty.

For convenience, here we assume that
(22)$$P(t)=\sum_{t=0}^{T-1}P_{a}(t)$$

Hence, P3 can be expressed as:(23)P3: minΔΘ(t)+VE(P(t)|Θ(t))
s.t. (11), (12), (13)
where *V* denotes the weight of the objective function, ΔΘ(t) denotes the drift of the queue, i.e., the stability performance of the queue.

Next, we show the specific form in preparation for P4 to find the upper bound. First, we show the proof of $A{\left(t+1\right)}^{2}-A{\left(t\right)}^{2}$.
$$\begin{array}{l} A(t+1{)}^{2}-A(t{)}^{2}\\ =\max(A(t)+\sum_{i=1}^{N}F_{i,j}(t)-F_{j},0{)}^{2}-A(t{)}^{2}\\ \leq 2A(t)(\sum_{i=1}^{N}F_{i,j}(t)-F_{j})+(\sum_{i=1}^{N}F_{i,j}(t)-F_{j}{)}^{2}\\ \leq 2A(t)\sum_{i=1}^{N}F_{i,j}(t)-2A(t)F_{j}+(\sum_{i=1}^{N}F_{i,j}(t){)}^{2}-2F_{j}\sum_{i=1}^{N}F_{i,j}(t)+F_{j}^{2}\\ \leq 2(A(t)-F_{j})\sum_{i=1}^{N}F_{i,j}(t)+(\sum_{i=1}^{N}F_{i,j}(t){)}^{2}+F_{j}^{2}\\ \leq 2(A(t)-F_{j})\sum_{i=1}^{N}F_{i,j}(t)+(N+1)F_{j}^{2}\\ =D_{1}+2(A(t)-F_{j})\sum_{i=1}^{N}F_{i,j}(t) \end{array}$$

Here, the value *D*_1_, which is fixed, can be expressed as
$$D_{1}=(N+1)F_{j}^{2}$$

Similarly, we can obtain
$$B{\left(t+1\right)}^{2}-B{\left(t\right)}^{2}=D_{2}+2(B(t)-C_{j})\sum_{i=1}^{N}C_{i,j}(t)$$

Here, the value *D*_2_, which is fixed, can be expressed as
$$D_{2}=(N+1)C_{j}^{2}$$

The proof of queue *C*’s drift is as follows.
$$\begin{array}{l} C(t+1{)}^{2}-C(t{)}^{2}\\ =\max(C(t)+\tau_{i}(t)+\tau_{c}(t)+\tau_{j}(t)-\tau ,0{)}^{2}-C(t{)}^{2}\\ \leq 2C(t)(\tau_{i}(t)+\tau_{c}(t)+\tau_{j}(t)-\tau )+(\tau_{i}(t)+\tau_{c}(t)+\tau_{j}(t)-\tau {)}^{2}\\ \leq 2C(t)(\tau_{i}(t)+\tau_{c}(t)+\tau_{j}(t))+(\tau_{i}(t)+\tau_{c}(t)+\tau_{j}(t){)}^{2}+\tau^{2}\\ =D_{3}+2C(t)(\tau_{i}(t)+\tau_{c}(t)+\tau_{j}(t))+(\tau_{i}(t)+\tau_{c}(t)+\tau_{j}(t){)}^{2} \end{array}$$

Here, the value *D_3_*, which is fixed, can be expressed as $\tau^{2}$.

Therefore, the P3 problem can be expanded to generate the P4 problem as follows:(24)$$\begin{array}{l} P4:\;\min VP(t)+(A(t)-F_{j})\times \sum_{i=1}^{N}F_{i,j}(t)+(B(t)-C_{j})\sum_{i=1}^{N}C_{i,j}(t)\\ +\frac{1}{2}(\tau_{i}(t)+\tau_{c}(t)+\tau_{j}(t))\times (2C(t)+\tau_{i}(t)+\tau_{c}(t)+\tau_{j}(t)) \end{array}$$
s.t. (13), (14), (15)

Assume that P4 has an upper bound, and its upper bound is the constant *C*. First, we have to prove that virtual queue *A*, virtual queue *B*, and virtual queue *C* are stable. The proof is as follows.
(25)$$\begin{array}{ll} L(\Theta (t+1))-L(\Theta (t))+VP(t) & \leq C\\ L(\Theta (t+1))-L(\Theta (t)) & \leq C\\ L(\Theta (T))-L(\Theta (0)) & \leq TC \end{array}$$

Let L(Θ(0))=0, then we can get
(26)$$\begin{array}{cc} \frac{1}{2}(A(t{)}^{2}+B(t{)}^{2}+C(t{)}^{2}) & \leq TC\\ A(t{)}^{2} & \leq 2TC\\ A(t) & \leq \sqrt{2TC} \end{array}$$

Therefore,
$$\underset{T\to \infty }{\lim}\frac{A(T)}{T}\leq \underset{T\to \infty }{\lim}\frac{\sqrt{2TC}}{T}=0$$

Finally, we can get
$$\underset{T\to \infty }{\lim}\frac{A(T)}{T}=0$$

Similarly,
$$\begin{array}{c} \underset{T\to \infty }{\lim}\frac{B(T)}{T}=0\\ \underset{T\to \infty }{\lim}\frac{C(T)}{T}=0 \end{array}$$

Therefore, its queues are stable, so it is possible to use an approximate analysis to analyze the upper bound of P4.

### 4.3. Proof of the Upper-Bound Calculation

Previously, we proved that P4 was upper-bounded, so next, we try to find its specific value. First, substituting the computational drift into the formula, we prove that
(27)$$\underset{T\to \infty }{\lim}\frac{1}{T}\sum_{t=0}^{T-1}P^{\prime }(t)\leq P^{*}+\zeta$$
where
(28)$$\zeta =\frac{(N^{2}+1)(F_{j}^{2}+C_{j}^{2})+2\tau^{2}+2NA_{\max}F_{j}+2NB_{\max}C_{j}}{2V}$$
is the upper bound of P4, and the Lyapunov drift-plus-penalty [21] enables the problem to be solved optimally within a certain range of deviations.

The proof is as follows.
$$\begin{array}{l} \Delta \Theta^{\prime }(t)+VE(P^{\prime }(t)|\Theta (t))\\ \leq \frac{D_{1}+D_{2}+D_{3}}{2}+VP^{\prime }(t)+(A(t)-F_{j})\sum_{i=1}^{N}{F^{\prime }}_{i,j}(t)\\ +(B(t)-C_{j})\sum_{i=1}^{N}{C^{\prime }}_{i,j}(t)+\frac{1}{2}({\tau_{i}}^{\prime }(t)+{\tau_{c}}^{\prime }(t)\\ +{\tau_{j}}^{\prime }(t))(2C(t)+{\tau_{i}}^{\prime }(t)+{\tau_{c}}^{\prime }(t)+{\tau_{j}}^{\prime }(t))\\ \leq \frac{D_{1}+D_{2}{+D}_{3}}{2}+VP^{*}+N(A_{\max}F_{j}+B_{\max}C_{j})\\ +\frac{1}{2}({\tau_{i}}^{\prime }(t)+{\tau_{c}}^{\prime }(t)+{\tau_{j}}^{\prime }(t))(2C(t)+{\tau_{i}}^{\prime }(t)\\ +{\tau_{c}}^{\prime }(t)+{\tau_{j}}^{\prime }(t))\\ \leq \frac{D_{1}+D_{2}{+D}_{3}}{2}+VP^{*}+2C_{\max}\tau +\tau^{2}\\ +N(A_{\max}F_{j}+B_{\max}C_{j}) \end{array}$$

By moving $\Delta \Theta^{\prime }(t)$ to the right and dividing by *V* at the same time, the original expression can be converted to the following inequality.
$$\begin{array}{l} \underset{T\to \infty }{\lim}\frac{1}{T}\sum_{t=0}^{T-1}P^{\prime }(t)\\ \leq \frac{D_{1}+D_{2}{+D}_{3}}{2V}+P^{*}+\frac{2\tau C_{\max}}{V}+\frac{\tau^{2}}{V}\\ +\frac{N(A_{\max}F_{j}+B_{\max}C_{j})}{V}-\frac{\Delta \Theta^{\prime }(t)}{V} \end{array}$$

Next, we unfold $\Delta \Theta^{\prime }(t)$ to obtain the upper bound ζ
$$\begin{array}{l} \underset{T\to \infty }{\lim}\frac{1}{T}\sum_{t=0}^{T-1}P^{\prime }(t)\\ \leq \frac{D_{1}+D_{2}{+D}_{3}}{2V}+P^{*}+\frac{2\tau C_{\max}}{V}+\frac{\tau^{2}}{V}+\frac{N(A_{\max}F_{j}+B_{\max}C_{j})}{V}-\frac{\Delta \Theta^{\prime }(t)}{V}\\ =\frac{D_{1}+D_{2}{+D}_{3}}{2V}+P^{*}+\frac{2\tau C_{\max}}{V}+\frac{\tau^{2}}{V}+\frac{N(A_{\max}F_{j}+B_{\max}C_{j})}{V}-\underset{T\to \infty }{\lim}\frac{L(\Theta^{\prime }(T-1))-L(\Theta^{\prime }(0))}{VT}\\ =P^{*}+\frac{{\left(N+1\right)}^{2}F_{j}^{2}+{\left(N+1\right)}^{2}C_{j}^{2}+\tau^{2}}{2V}+\frac{NA_{\max}F_{j}}{V}+\frac{NB_{\max}C_{j}}{V}+\frac{\tau^{2}}{2V}\\ =P^{*}+\frac{(N^{2}+1)(F_{j}^{2}+C_{j}^{2})+2\tau^{2}+2NA_{\max}F_{j}+2NB_{\max}C_{j}}{2V} \end{array}$$

The proof is complete.

## 5. Algorithm Design

Based on the theoretical proof, we designed the PSO algorithm under the Lyapunov framework (LPSO) and used Algorithms 1 and 2 to solve P4.**Algorithm 1** LPSO for Task Type 1
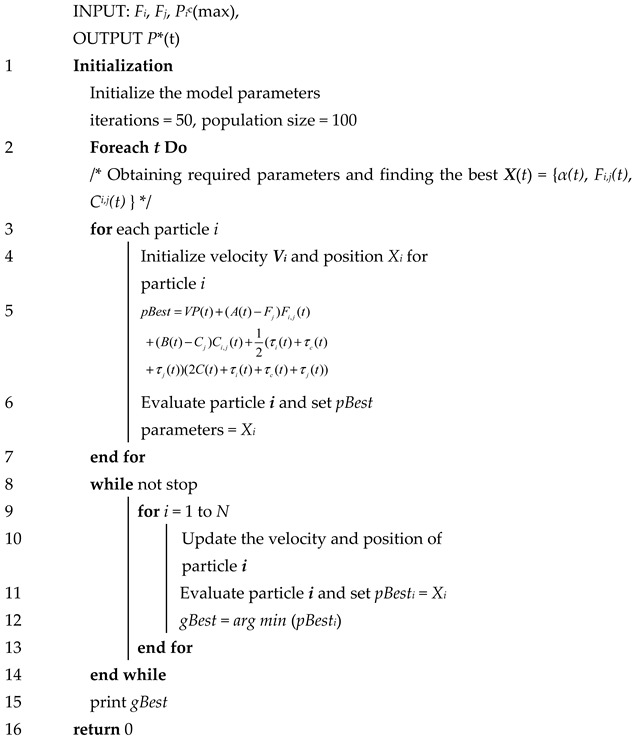


Line 1: Initialize the required parameters in the IoT device, such as the controlling parameter *V*, iterations, population size.

Line 3–7: For all tasks, calculate the value of *P** by node.

Line 8–16: Compare each calculated value and choose the minimum value to update the queue until it remains stable.**Algorithm 2** LPSO for Task Type 2
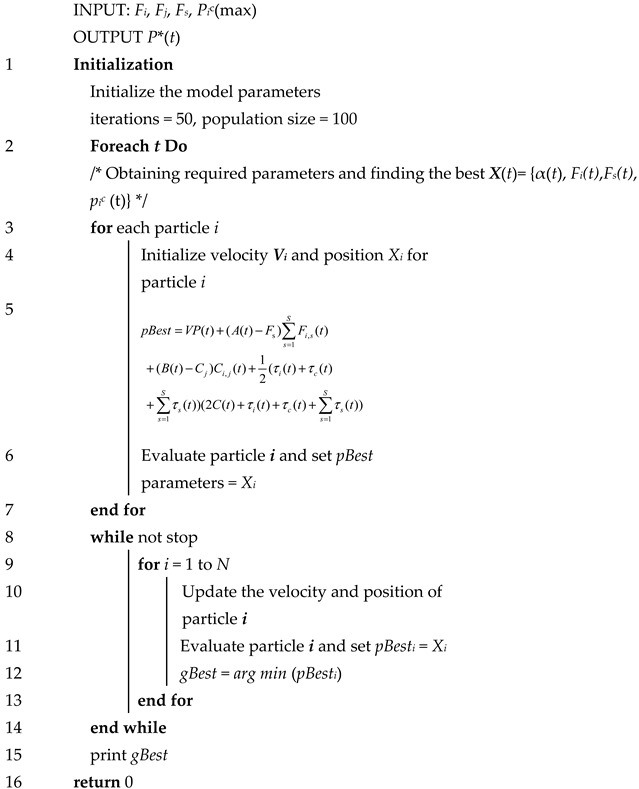


Line 1: Initialize the required parameters in the IoT device, such as the controlling parameter *V*, iterations, population size.

Line 3–7: For all tasks, calculate the value of *P** by the pro-node

Line 8–16: Compare each calculated value and choose the minimum value to update the queue until it remains stable.

The difference between Algorithms 1 and 2 is that Task 2 requires scheduling more nodes to perform the computation.

## 6. Numerical Results

In this section we simulated the construction of a fog environment consisting of *N* = {1, 5, 10} and *M* = 2. The *N* IoT nodes were randomly distributed around the coverage area of the fog nodes. We used the experimental setup of [8] with a channel bandwidth of *W* = 5 MHz and background noise *N*_0_ = −100 dm. The specific parameters in the experiments are shown in Table 2. The configuration of our simulation machine was an Intel Core i7-10200H CPU, RAM 32 GB system with Windows 10. The simulation platform used in this article was EdgeCloudSim.

We studied the relationship between the number of different IoT nodes and the energy consumption in Figure 2. Here, the LPSO parameters we chose to calculate are the task standard deviation $\sigma_{i}{=D}_{i}$, the weight *V* = 50, the arrival rate $\lambda_{i}(t)=\mathrm{unif}\;[1,1.1]$. The average energy consumption grows with the number of IoT nodes, because as the number of nodes increases, the number of arrivals of random tasks also increases, thus the average energy consumption of the three algorithms also increases. With the same number of nodes, we can see that LPSO has the lowest average energy consumption, PSO has the middle energy consumption, and the greedy algorithm has the highest energy consumption. Since in the same situation, compared to PSO, LPSO increases the control of virtual queues (including channel virtual queues, computing resource virtual queues), the realization of LPSO is better than that of PSO.

We also studied the average energy consumption of LPSO for different weights V in Figure 3. The performance varies at different weights, with the highest energy consumption at *V* = 1 and the lowest energy consumption at *V* = 50, where *V* is the weight parameter for energy consumption, and a larger value of *V* means that the energy consumption value has a greater impact on the objective value. This is because when the value of *V* increases, a small decrease in energy consumption leads to a large decrease in the value of the objective function, and then the optimization process tends to reduce the value of the energy consumption, which leads to a lower value of the objective function. In the experiments, we considered the standard deviation $\sigma_{i}{=D}_{i}$ and the arrival rate $\lambda_{i}(t)=\mathrm{unif}\;[1,1.1]$.

Figure 4 and Figure 5 show the comparison between LPSO, PSO, and the greedy algorithm for the same number of nodes and other cases, including the total energy consumption and average energy consumption, where our proposed LPSO outperforms the other two. Because the measure taken by the greedy algorithm is to complete all task requests as much as possible, the power of all devices is set to the maximum based on the greedy algorithm. PSO is more reasonable for resource allocation than the greedy algorithm. Compared with the greedy algorithm, PSO pays more attention to the rationality of resource utilization. On the basis of PSO, LPSO increases the control of computing resources and channel resources, thus LPSO performs better than PSO and the greedy algorithm.

In addition, we also compared the energy consumption of LPSO for different arrival rates, as shown in Figure 6, and as expected, the higher the value of the arrival rate $\lambda_{i}(t)$, the more energy is consumed. Here we set the weight *V* = 50 because LPSO performs best at *V* = 50. A higher arrival rate leads to more computational tasks competing for the limited network and computing resources, thus increasing the queuing time of the computing workload, which requires more computing resources to ensure that it can be completed within the specified time.

Finally, Figure 7 shows the completion degree of different algorithms under different arrival rates. When all conditions are the same, including the number of nodes, task size, and task delay, we can compare LPSO and the other algorithms in the number of tasks completed at different task arrival rates. LPSO performs better than PSO because PSO only considers the most ideal resource allocation scenario (i.e., each task just has to match its requirements), which causes the queue to block. This leads to an increase in queuing time, and LPSO combines the optimization of computing resources and channel resources in the case of a long time, where every time slot maximizes the use of resources while keeping the queue stable. Moreover, the reason why the greedy algorithm does not complete tasks on time is because it completes as many tasks as possible; in other words, all the equipment power is turned on to the maximum, and the energy consumption is exchanged for the number of completed tasks.

## 7. Conclusions

The Lyapunov-based PSO algorithm we designed achieved a balanced distribution of transmission power and computing resources through the control of channel resources and computing resources and minimized the energy consumption value while ensuring that as many tasks as possible were completed. Comparing LPSO and PSO, our algorithm was better than the original PSO algorithm and the greedy algorithm in the performance index of energy consumption required to complete the task. The number of tasks completed was slightly lower than that of the greedy algorithm, because the greedy algorithm used a larger amount of energy than the LPSO algorithm to achieve its result, which obviously outweighed the gains. Regarding future work, we will set up nodes without mobility and fog nodes without energy constraints, as some tasks may not be completed if there are energy constraints. We hope to increase these constraints in future work, and further increase the number of tasks completed. Furthermore, regarding the task decomposition, since some subtasks might have a sequential relationship of backward and forward processing, we consider using a directed acyclic graph (DAG) in future work.

## Figures and Tables

**Figure 1 sensors-22-03527-f001:**
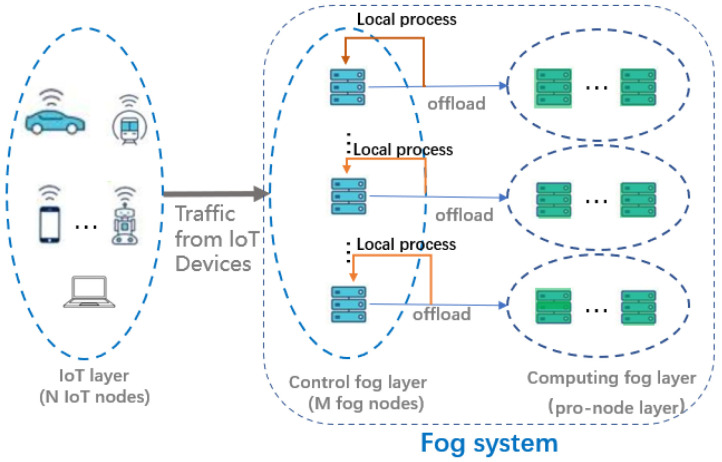
System model diagram indicating whether the task that comes from the IoT device is processed locally or handed over to a pro-node for processing, mainly determined by the resources required by the task. Every pro-node has *S* calculation subnodes.

**Figure 2 sensors-22-03527-f002:**
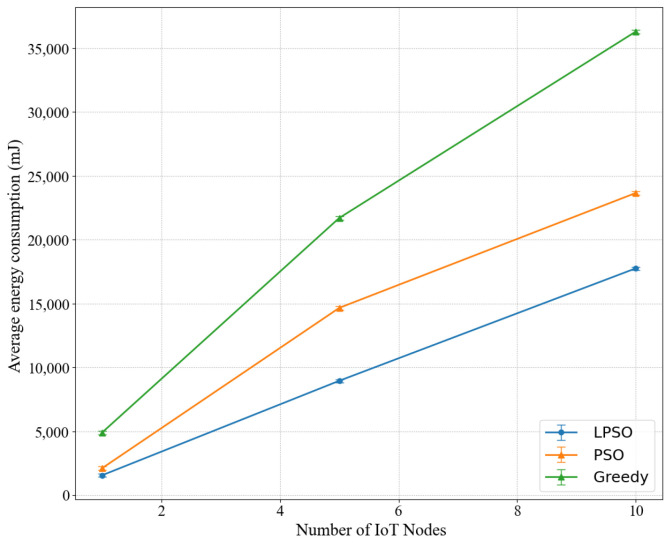
Influence of IoT equipment quantity on energy consumption.

**Figure 3 sensors-22-03527-f003:**
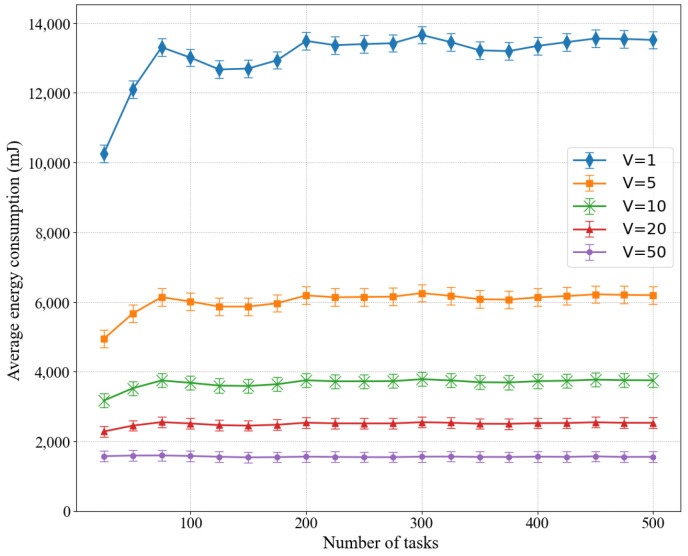
Influence of different values of weight *V* on average energy consumption.

**Figure 4 sensors-22-03527-f004:**
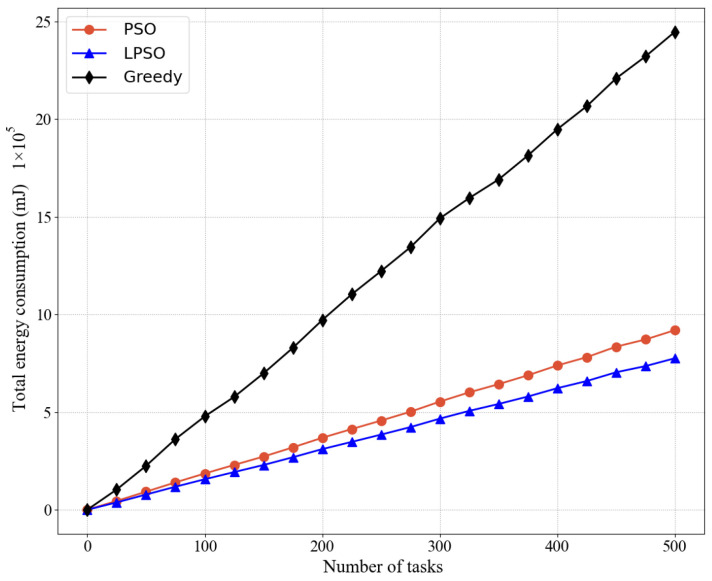
Comparing the total energy consumption of different algorithms in the same case where the weight of LPSO is *V* = 50.

**Figure 5 sensors-22-03527-f005:**
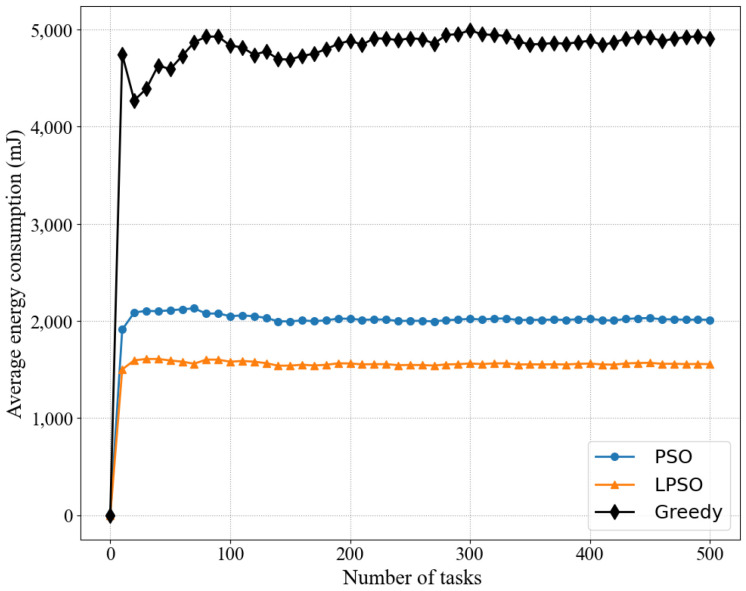
Comparing the average energy consumption of different algorithms in the same case where the weight of LPSO is *V* = 50.

**Figure 6 sensors-22-03527-f006:**
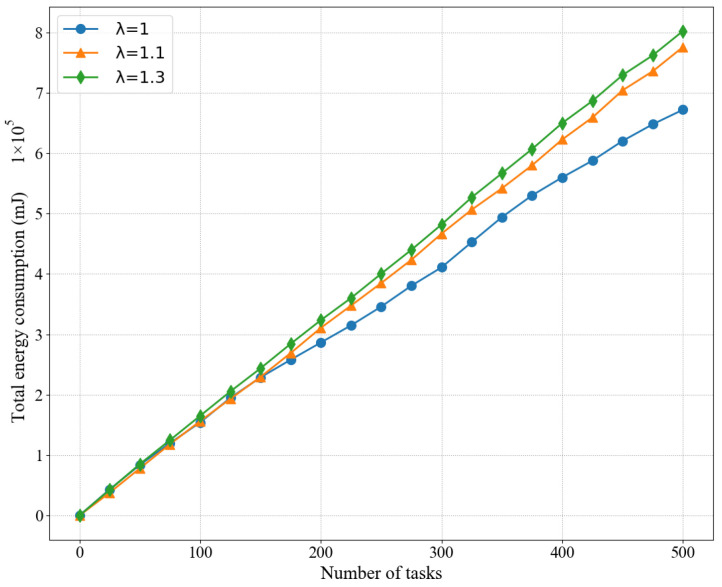
Influence of different arrival rates on the average energy consumption of the LPSO algorithm.

**Figure 7 sensors-22-03527-f007:**
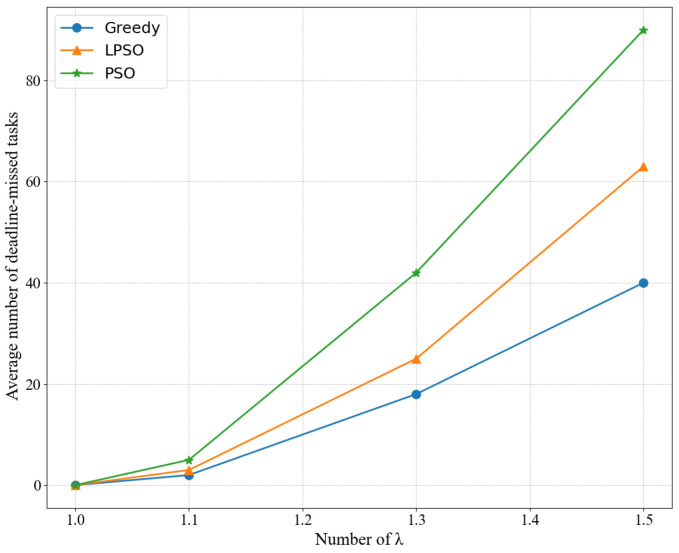
Completion degree of different algorithms under different arrival rates.

**Table 1 sensors-22-03527-t001:** Defitions of notations.

Notations	Meanings
*F_i_*, *F_j_*, *F_s_*	Computing power of devices *i*, *j*, *s*, respectively
*M*	Set of IoT nodes
*N*	Set of fog nodes
*S*	Set of computing devices in pro-node
*F_i,j_(t)*	Computing power allocated to *i* by device *j* in time slot t
*F_i,s_(t)*	Computing power allocated to *i* by device *s* in time slot t
*τ_i_*, *τ_j_*, *τ_s_*	Node *i*, *j*, *s*, calculation delay
*τ_c_*	Transmission delay
*τ(t)*	Delay at time slot t
*τ*	Target delay vector
*P_c_*	Sent tasks
*λ_i_(t)*	Arrival rate of tasks for time slot *t*
*σ*	Device *i* computes the standard deviation of the task size
*D_i_*	Mean value of task size for device *i*
*C_j_*	Device *j*’s downstream bandwidth
*Ls*	Local processing task delay
*P_a_(t)*	Total energy consumption of time slot t
*P_i_(t)*	Energy consumed by fog node *i*
*P_j_(t)*	Energy consumed by fog node *j*
*P_s_(t)*	Energy consumed by fog node *s*
*P_i,j_^c^*	Energy consumption of device *i* to transfer tasks to *j*
*P_i_^c^*	Transmit power of device *i*
*C_i,j_*	Device *i* to *j* transfer rate
*α*	Offload rate
*P^b^_i_*	Device *i*’s battery

**Table 2 sensors-22-03527-t002:** Parameter settings in the experiments.

Parameter	Value
*N*	1, 5, 10
*W*	5 MHz
*N_0_*	−100 dBm
*λ_i_(t)*	1, 1.1, 1.3
*D_i_*	unif [100, 300] KB
task_delay	unif [8, 15] ms
*σ_i_*	*D_i_*, *5D_i_*, *7D_i_*
*F_i_*	500 MHz
*F_j_*	1 GHz
*F_s_*	800 MHz
*p_i_^c^*	200 W
*V*	1, 5, 10, 20, 50
ς	10^−10^

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
