# Peer review of "Resource Scheduling and Energy Consumption Optimization Based on Lyapunov Optimization in Fog Computing"

_sensors, 2022, doi:10.3390/s22093527_

Round 1

Reviewer 1 Report

This paper presents a resource scheduling and energy consumption optimization scheme based on Particle Swarm Optimization (PSO), integrating concepts of Lyapunov optimization. The resulting scheme is called LPSO. The paper is interesting, but the explanation requires a lot of improvement in order to deserve publication. My main remarks are the following:

1) P. 2: This is a minor one. In "improve the transmission rate when the task delay requirement is high, or reduce the transmission rate when the task delay is low to achieve the purpose of energy saving", shouldn't it be the other way around? When the required delay is LOW is when the transmission rate must increase, and vice versa. 

2) Table 1: Not all math symbols are listed. Some are not even introduced in the text. For example, what is N in Eq. (2), or M in Eq. (5)? What is the S in Eq. (6)? Also in Eq. (6), what is the meaning of Ps (by the way, Ps in mentioned just after Eq. (5) without being used there), Pi and Pj (by the way, there is no Pj in Eq. (6), though it is referred in the text just after that equation)? Pa(t) appears in Eq. (7), but is never defined. 

3) P. 6: it is written that "the decision variables are influenced by the size of tasks". Where is this size being taken into account in the equations?

4) P.6: "The size of the channel bandwidth occupied by the task should be
smaller than the total channel bandwidth in the long term, due to the limitation of the device i's own transmission speed." Will different tasks share the communication channel? Where is that being taken into account? Is it Eq. (9)? But C_i,j is defined in Table 1 simply as the "Transfer rate of devices i to j". By the way, C_i,j seems to be repeated in Table 1: i,j are subscripts in one instance, and superscripts in the other. Are those variables the same?

5) The Lyapunov virtual queues expressed in Eq. (14a), Eq. (14b), and Eq. (14c) should be better explained. 

6) Regarding Eq. (14c) in particular, how does C(t) take into account bi-directional communication. Eq. (14c) seems to consider only unidirectional communication. Why 2*tau(t)?

7) Regarding the simulations, is the simulator a discrete event simulator (DES), or are the results the curves provided by the optimization model? In my opinion, the evaluation should be based on a DES.

8) This is probably related with the previous remark. Curves should be presented for the successful task completion rate (i.e., tasks completed within the deadline).

9) The paper still has a lot of typos, misplaced references to variables (e.g., variables not appearing in an equation are referred as if they appeared), uppercase/lowercase issues with variables, and what appears to be some inconsistency in the naming of some variables. A careful revision of the text is a must.

Reviewer 2 Report

This paper presents a heuristic particle swarm optimization algorithm based on Lyapunov framework (LPSO) for resource scheduling and energy consumption optimization in fog computing.

In the last paragraph of the introduction section, “Chapter” should be “Section”.

Table 1 is basically the nomenclature. It would be better to put these symbols under nomenclature.

In Figure 1, the sets of N (IoT nodes) and M (fog nodes) should be labelled.

The optimisation problem in Eq(7) is not properly expressed. You need to add “min” and mention the decision variables.

How are A, B, and C in Eq(14) are initialised? Some clarifications should be given.

In the optimisation algorithm 1 and 2, the algorithm output is P*(t). What about the decision variables? The decision variables for the optimal solution should also be given.

Some details on implementing the PSO should be given, such as population size and stopping criterion. It would be better to add a graph to show the convergence over the generations.

Reviewer 3 Report

This paper proposes a Particle Swarm Optimization (PSO) algorithm based on LPSO to minimize the energy consumption for delay-sensitive IoT tasks. A key novelty is the consideration for computational energy consumption of IoT nodes, transmission energy consumption and fog node computing energy consumption. With a simulation study, the proposed algorithm was compared with the Greedy algorithm and the naive PSO algorithm. 

While the proposed algorithm seems sound, its evaluation and the discussion of the implications of the results are weak. 

As for the evaluation, the author should use a proper simulation tool (validated and approved by the academia/research community). For example, please check the tools of https://github.com/Cloudslab Otherwise, it is hard to accept the credibility of the simulations.  

A discussion section should be included. The authors should discuss any threats to validity. 

Please expand the conclusion section too.  

The related work only includes a few related works. There are many works on energy consumption optimization.  

Round 2

Reviewer 1 Report

The authors have made an effort to improve the paper, but the result is still not satisfying to allow publication. Regarding the quality of the English text, it still requires a major revision. Even some of the added paragraphs are not correct. For example, in P. 5, in yellow: "Every pro-node with S calculation subnodes." This is not a proper sentence and does not even have a verb. The authors have also not paid attention to the consistent use of lowercase/uppercase characters. For example, LPSO is sometimes written Lpso (see Fig. 7). The authors must revise the text properly, or hire the services of a professional to do it.

However, there are more concerns, namely of scientific character:

1) The authors claim that bi-directional communication does not apply because only the results are transmitted. However, in order to generate the results, the computing nodes must first receive the job requests. I don't see how the communication of job requests can be ignored.

2) The authors have added some results of task completion rate in Fig. 7. In this figure, it is obvious that PSO performs significantly better than LPSO. However, no comment or explanation is provided. It is now obvious that LPSO is more energy-efficient by sacrificing the task completion rate. This important conclusion really requires a justification and explanation about the kinds of scenarios where LPSO can be used. Is there a way to tune LPSO, so that it can trade-off energy consumption for a better task completion rate?

3) In networking related research, it is a good practice to validate the analytical models using discrete event simulation. This is not currently done in this paper. 

Reviewer 2 Report

The authors have adequately addressed my comments and the revised manuscript can be accepted.

Author Response

Thanks for your submission, and I will continue to work on it.

Reviewer 3 Report

The authors have addressed my comments for the previous version of the paper. However, a detailed response for the comments would have been good.

Round 3

Reviewer 1 Report

In general, I am now satisfied with the explanation and changes introduced by the authors. The English text can still be improved. For example: " This will lead to an increase in queuing time, and LPSO combines the optimization of computing resources and channel resources, in the case of a long time, every time slot maximizes the use of resources while keeping the queue stable." This long sentence can be divided in three separate short sentences, making it much easier to read and understand.